# Analysis of China's Coastline Changes during 1990–2020

**Kaixin Li** [1,2,3], **Li Zhang** [1,2,*], **Bowei Chen** [1,2] , **Jian Zuo** [1,2], **Fan Yang** [1,2] and **Li Li** [3]

1    International Research Center of Big Data for Sustainable Development Goals, Beijing 100094, China
2    Key Laboratory of Digital Earth Science, Aerospace Information Research Institute,
     Chinese Academy of Sciences, Beijing 100094, China
3    School of Marine Technology and Geomatics, Jiangsu Ocean University, Lianyungang 222005, China
*    Correspondence: zhangli@aircas.ac.cn; Tel.: +86-186-0013-2968

**Abstract:** As the boundary between the sea and the land, information on the location and type of coastline constantly changes with environmental changes in coastal zones. Monitoring of coastline changes in long time series becomes important for the monitoring and assessment of the coastal zone environment. In this study, Landsat series images from five time periods (1990, 2000, 2010, 2015, and 2020) were selected for monitoring and analyzing the changes in coastline length, sea–land pattern, the index of coastline diversity, and fractal dimension characteristics. Our conclusions are as follows: (1) The lengths of the entire coastline and the artificial coastline of mainland China increased from 30,041.22 km and 10,022.49 km in 1990 to 32,977.34 km and 17,660.84 km in 2020, with annual change rates of 97.87 km/year and 254.61 km/year, respectively. From 1990 to 2020, the rate of natural coastline decreased from 66.68% to 42.29%, and the artificial coastline increased from 33.32% to 57.71%. (2) The length of natural coastline decreased from 20,018.73 km to 15,316.5 km; among the types of natural coastline, the length of sandy coastline and bedrock coastline decreased the most, at 2062.95 km and 1815.8 km, respectively. (3) The coastal zone of mainland China had a significant increase in land area, with a net increase of about 10,902.55 km$^2$. (4) The index of coastline diversity continued to decrease, and the coastline structure tended to be simple. The fractal dimension of the mainland coastline was consistent with the trend of the length of the coastline, which basically shows an increasing trend. Therefore, the length of the mainland coastline and artificial coastline displayed an upward trend between 1990 and 2020, which also led to simpler coastline diversity and more complex coastline shapes. Since the first year of the SDGs (2015), the growth rate of the artificial coastline has decreased by 158.32 km/year compared with that between 2010 and 2015. In recent years, China has enacted a number of laws, regulations, and action plans to protect its coastline, and it has proposed that by 2020, the proportion of natural coastline will be no less than 35%. The rapid development of China's coastal areas drives the construction of coastal zone cities but also creates a variety of challenges for the ecological environment of the coastal zone, and the management and sustainable use of the mainland coastline resources should be further strengthened.

**Keywords:** mainland China; coastline; landsat; index of coastline type diversity; fractal dimensions





## 1. Introduction

The coastal zone is a sensitive area where the ocean, land, and atmosphere interact and intersect with many constituent elements [1]. It is the center of the world's economic activities and the main site of development and construction for all countries [2]. Managing and protecting the coastal zone ecosystem is of great significance to the sustainable development of humanity [1]. However, the coastal zone is ecologically fragile [3], and human activities, while driving economic development, have also had negative impacts on the coastal zone, such as the degradation of coastal wetlands and ecosystem service functions.

The coastline, as one of the important elements of the coastal zone, contains rich marine resources and environmental information while dividing the sea from the land [4–6]. Information on the coastline's position and type is constantly changing in response to changes

in the coastal zone environment [7]. With the dramatic impact of global climate change and various natural phenomena, such as erosion, saltwater intrusion, subsidence, tsunamis and floods, and tropical cyclones, and human activities such as fish pond farming, land reclamation, and large engineering structures [8], monitoring of coastline changes has attracted the attention of a wide range of researchers.

The global coastal zone is affected by sea level rise and frequent human activities, and environmental problems are becoming more serious [9–11]. Through the analysis of the spatio-temporal pattern of coastal reclamation areas in the major deltas of Asia in different periods since 1990, Liu et al. found that the reclamation activities are the main driving force for changes in the length of the coastline and its advance toward the sea [12]. Approximately 12% of Southeast Asian islands have undergone changes in coastlines, resulting in a decrease of about 251 km$^2$ in area between 1990 and 2015 [13]. Approximately 22% of Australia's non-rocky coastline has retreated or grown significantly since 1988, with 16% changing by more than 0.5 m per year [14]. Over 1987–2017, 58% (50.7 km) of the coastline of the Nigerian Transgressive Mahin Mud Coast experienced retreat, and this rapid rate of coastline retreat has triggered land loss of 10.64 km$^2$ to the Atlantic [15]. A study by Hannes et al. suggests a sea level rise of 1.04 m in the worst-case scenario, which would threaten an area of 2840.64 km$^2$ in Colombia [16]. Based on the estimated value of future global sea level rise and the data of regional sea level rise in China around 1995, the estimated value of future relative sea level rise rate in some main coastal areas of China was 5–8 mm/year, and the relative sea level rise in the coastal zone of China may reach 30–45 cm by 2050 [17]. More than 68% of the coastlines in China tended to expand toward the sea, with a mean weighted linear regression rate of 24.30 m/year from the 1940s to 2014 [5]. The total area of reclamation along China's mainland coast was 11,162.89 km$^2$ between 1979 and 2014, and Shandong Province showed the largest reclamation area, which had reached 2736.54 km$^2$ [18].

With the development of remote sensing data, different methods have been developed to identify and analyze coastlines. Many studies use Landsat satellites due to their longest time series and medium resolutions. Using Landsat images, previous studies analyzed the coastline changes in the Pearl River Estuary [19], in the Bohai Sea [20], in the Yangtze River Estuary [21], and in the Zhoushan archipelago [22] of China; studied the coastline changes along the coast between Kanyakumari and Tuticorin of south India [23]; and studied coastline change in relation to meteorological conditions and human activities at Cape Cam Mau, Vietnam, 1990–2010 [24]. For localized areas, higher resolution satellite data are also used, such as aerial photographs, SPOT-5, QuickBird, WorldView 2/3, and Sentinel-2. For example, with historical aerial photographs and SPOT-5 data, Valderrama-Landeros assessed the coastal erosion and accretion trends along the coastline of the San Pedro River and the Santiago River on the Pacific coast of Mexico [25]. Dai et al. obtained 2-m resolution shoreline data by integrating QuickBird, WorldView-2, and WorldView-3 multispectral imagery [26]. Saleem and Awange extracted the coastlines of Liberia and Somalia for 2015–2018 using Sentinel-2 and Landsat 8 satellite imagery [27]. Assisted by other datasets, such as DEM, researchers analyzed the coastline of the north coast of Florida [28] and modeled coastline change rates in a Mediterranean coastal area [29].

China has become the world's second-largest economy in the course of decades of development, and its coastal zone is one of the fastest-growing and most economically dynamic regions in the world. The coastal provinces and cities occupy more than 70% of the country's large and medium-sized cities and nearly 40% of its population, generating more than 61% of the country's gross national product [30]. Along with the rapid development of China's coastal zone, its coastline has undergone dramatic changes during these decades, with human activities being the main factor behind the dramatic coastline changes. Since the early 20th century to 2014, the structure of China's mainland coastline has changed significantly, with the proportion of artificial coastline increasing to 67% [5].

In response to SDG 14.5, China introduced the *Management Measures for Coastline Protection and Utilization* in 2017 (State Oceanic Administration) and proposed that, by

2020, China's natural coastline retention rate would be no less than 35%. It is necessary to monitor coastline changes to achieve the sustainable development of coastal zone resources. Previous studies [14,19–21] had used Landsat series satellite images to extract coastlines, but long-term series coastline monitoring studies for China have only covered the period up to 2014 [5]. Coastline monitoring studies have not been updated since the first year of the Sustainable Development Goals (SDGs), 2015. To explore how the changes in the coastline have been affected by the implementation of a series of coastal zone protection measures in China after the first year of the SDGs (2015) and for a long-term of 30 years (1990–2020), we studied the coastline changes along the mainland of China (including seven islands that are heavily affected by human activities, such as Chongming Island, Kinmen Island, and Zhoushan Island) that had occurred by 1990, 2000, 2010, 2015, and 2020. Using Landsat images, we monitored and analyzed the changes in the coastline (including the position and types of coastline) over 30 years, which will provide a scientific basis for the planning of environmental protection measures and the sustainable development of coastal resources in China's coastal zone.

## 2. Coastal Zone of China

The coastal zone of China spans a large geographical area from north to south, with complex topography and huge climatic differences. With Hangzhou Bay as the boundary, the northern coastal zone is distributed with low hills and plains, most of which are plains; the southern coastal zone is dominated by low hills and tableland, with numerous islands and occasionally distributed estuary delta plains [31]. Influenced by the East Asian monsoon, the coastal zone area is hot and rainy in summer and cold and rainy in winter, with the high-temperature period coinciding with the rainy period. The temperature difference between north and south is large in winter and small in summer, and the seasonal distribution of precipitation is uneven. The spatial distribution of the annual mean temperature and precipitation gradually increases from north to south. According to the spatial characteristics of China's coastal zone, its ecosystems are divided into shallow-water ecosystems, marine–land overland ecosystems, terrestrial ecosystems, and terrestrial–aquatic ecosystems. They provide supply, regulation, support, and cultural services for human production and development [32].

The coastline along the mainland of China extends from the mouth of the Yalu River on the China–North Korea border to the north of the Beilun River on the China–Vietnam border in the south. It covers 12 major provinces, municipalities, autonomous regions, and special administrative regions, including Liaoning, Hebei, Tianjin, Shandong, Jiangsu, Shanghai, Zhejiang, Fujian, Guangdong, Hong Kong, Macao, and Guangxi provinces (Figure 1). About 42% of the population and 60% of the GDP of China are concentrated in these provinces and cities.

Considering that human activities have also had a great impact on the coastlines of nearby islands, this study provides a detailed classification of the coastlines of seven islands, namely, Donghai Island (Guangdong), Dongshan Island, Kinmen Island, and Xiamen Island (Fujian), Chongming Island (Shanghai), Zhoushan Island (Zhejiang), and Lantau Island (Hong Kong).

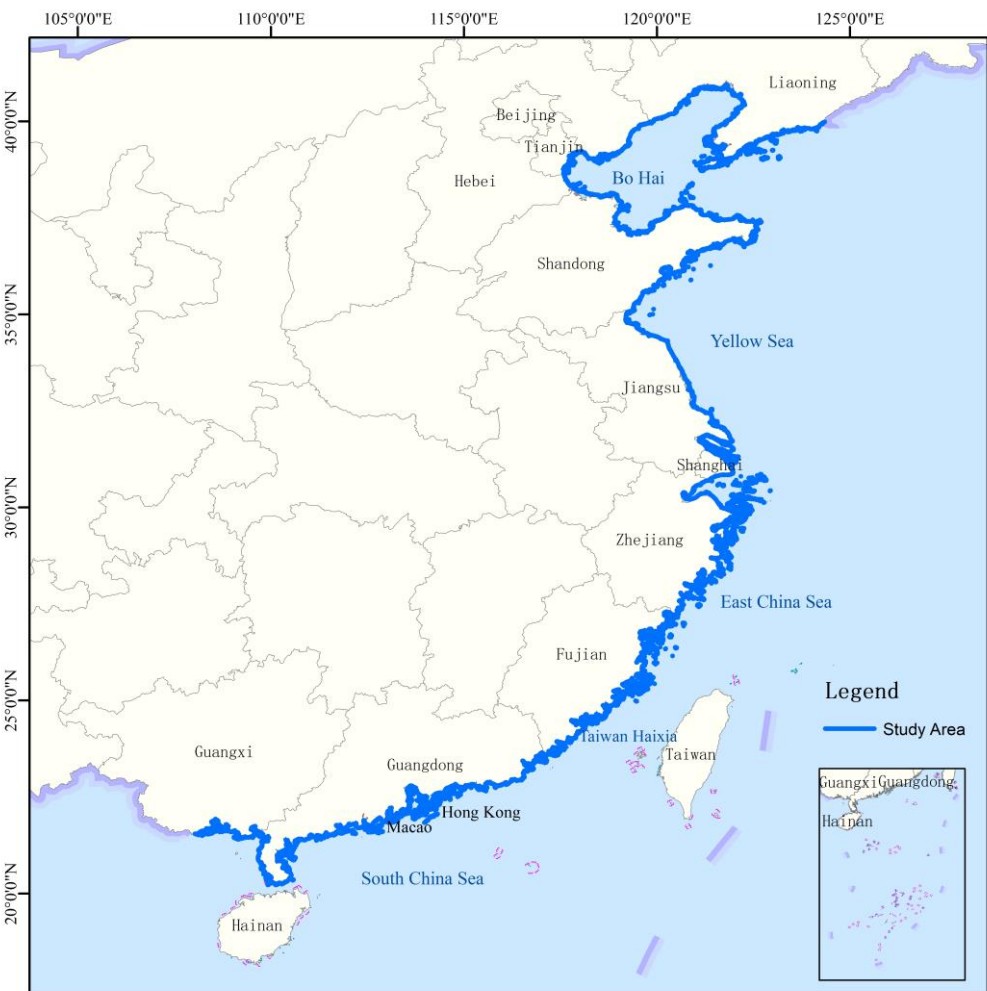

**Figure 1.** Geographic location of the study area.

## 3. Methods

### 3.1. System of Coastline Types

For the classification of coastlines, there is no globally accepted and authoritative unified classification system. According to the classification types defined by many researchers' classification experiences [5,6,33,34], we classified the coastline into 7 types: artificial coastline, estuary coastline, bedrock coastline, sandy coastline, biological coastline, silty coastline, and island coastline. They could be incorporated into two classes: natural and artificial coastlines.

Natural coastline refers to the maintenance of the natural coastal properties, where the shape and properties of the coastline are not altered by human activities. It generally refers to the long-term interaction between sea and land and the formation of a spatial natural boundary between sea and land. In its spatial form, it has a zigzag morphology, natural direction, relatively fixed location, ecological system structure integrity, functional stability, and other characteristics. According to the substrate characteristics of the intertidal zone where the coastline is located and the spatial morphology of the coastline, the natural coastline can be further divided into bedrock, sandy, silty, biological, and estuarine conditions [35]. Among them, sandy coastlines are one of the most common natural coastlines and are greatly influenced by human activities. Silty coastlines are often inextricably linked to biological coastlines and interact with each other. Biological coastlines, mainly mangrove forests, are commonly found near estuarine coastlines and silty coastlines, which are more stable but also more affected by human activities.

Artificial coastlines are built at the junction of land and sea for production and living needs and are mostly constructed with cement, stone, sand, concrete, and other construction materials, including breeding berms, salt field berms, docks (ports, fishing ports), jetties (tidal dikes, anti-slope dikes, shore protection barriers, etc.), transportation berms, energy extraction facilities, recreational facilities, artificial island construction, and other types. Most artificial coastlines are located in the intertidal zone or even the subtidal zone. The construction of artificial coastlines leads to a significant compression of the natural tidal space of the coast, or even a complete absence of it, damage to the structure of the ecosystem, and decay of the tidal wetland function.

In this study, different types of island coastlines were classified according to different levels of frequent human activities. For islands with no or fewer human activities, the types of coastlines were classified as island coastlines, which is a natural coastline. On the contrary, for islands (e.g., Zhoushan Island, Xiamen Island, and Chongming Island, et al.) with more frequent human activities, the types of coastline were classified in further detail, such as artificial, bedrock, and sandy, et al.

*3.2. Remote Sensing for Coastline Extraction*

Previous studies [14,19–21] had used Landsat series satellite images to extract coastlines, and it has been verified that the judgment error caused by the tide is generally less than 1 image pixel [36,37]. Therefore, the Landsat images meet the accuracy requirements for research. In this study, 180 cloud-free or low-cloud Landsat 5 TM and Landsat 8 OLI images with 30 m resolution were freely downloaded from the United States Geological Survey (USGS) (https://earthexplorer.usgs.gov, accessed on 1 March 2021). The images cover the years 1990, 2000, 2010, 2015, and 2020. When the cloud cover was high or the image quality was poor, data for 1–2 years near the cloud cover were selected as a supplement. To avoid the effects of tides, we selected images from April to October through the years.

Firstly, based on the remote sensing images filtered by cloud cover, a manual method was used to extract the marine and land boundaries as coastlines. The acquired coastlines were then classified by visual interpretation methods, as shown in Table 1. Due to the different mechanisms by which different coastlines are formed, the texture characteristics expressed in remote sensing images are also different. Based on the interpreted signals for coastlines in the Landsat satellite images, combined with field survey data, the image features are described according to the classification system used in this study, described in Table 1.

**Table 1.** Interpretation of coastline types.

| First Class | Secondary Class | Definition and Interpretation of Signs | Image Sample (Landsat8 OLI 762) |
|---|---|---|---|
| Natural | Estuary | Estuary coastlines are generally located at the mouth of the river into the sea and are the dividing lines between the river and the sea. They are distributed at the estuaries of rivers that enter the sea and have a width of more than 100 m. |  |
| | Bedrock | Bedrock coastlines are formed by geological activities and wave action and consist of hard, bare rocks with steep terrain. There are often prominent headlands and bays between the headlands that go deep into the land. Coastlines between headlands and bays are long and winding, and erosion and accumulation are intertwined, with erosion generally occurring at the headlands and accumulation occurring in the bays. |  |

**Table 1.** *Cont.*

| First Class | Secondary Class | Definition and Interpretation of Signs | Image Sample (Landsat8 OLI 762) |
|---|---|---|---|
| Natural | Sandy | Sandy coastlines are located in open bays before the bedrock headlands. Sandy coastlines are mainly composed of fine sand, chalk, and silt; beaches are wide; the coastlines are straight and long, and many sand dams, offshore dams, and other accumulation features have often been developed along them; and these coastlines are generally in a state of erosion and recession. |  |
| | Biological | Biological coastlines are a special kind of coastline developed by reef-building coral and mangroves in the coastal zone. There are lush mangrove communities around the biological coastline, forming a natural mangrove landscape landform that can weaken the flow of water and achieve the role of shore protection. Most of the biological coastlines on Hainan Island are dominated by mangrove coastlines, which grow in the upper intertidal zone. |  |
| | Silty | Silty shore grows in hidden bays, and the shore forms are mostly gentle silty beaches made of powdered sand, clay, plant humus, etc., and are mostly greenish gray or greenish black, with weak hydrodynamic conditions and generally tidal ditches. |  |
| | Island | Coastlines are further divided into island coastlines and mainland coastlines. The water–land boundary of an isolated island or atoll far from land is called the island coastline. |  |
| Artificial | | Artificial coastlines are coastlines that have changed from their original natural state and that were constructed entirely by artificial action. They are distributed in areas where there are harbors, docks, fish ponds, salt pans, tidal dikes, breakwaters, tidal gates, and other structures. |  |

*3.3. Analytical Method of Characteristics of Changes to Coastline*

3.3.1. Index of Coastline Utilization Degree (ICUD)

The index of coastline utilization degree indicates the degree to which the coastline is influenced by anthropogenic effects [38]. Referring to the calculation method of the comprehensive index of land use degree, this study assigns different human action intensity

indices to different types of coastlines according to the degree of human activity on the coastline: bedrock = estuary = 1, sandy = 2, biological = silty = 3, and artificial = 4, and the following formula is used for calculation [39]:

$$ICUD = \sum_{i=1}^{n} (A_i \times C_i) \times 100 \tag{1}$$

where $n$ represents the number of types of coastlines; $A_i$ represents the human action intensity index corresponding to the $i$-type coastline; and $C_i$ represents the percentage of the length of the $i$ type coastline. The larger the ICUD, the greater the impact of the coastline on human health.

### 3.3.2. Sea-Land Pattern

The area method uses the area between two coastlines to characterize the current state of coastline change. The net increase in land area of the coastal zone is calculated with the following equation:

$$S' = S^+ + S^- \tag{2}$$

where $S^+$ is the increase in area and $S^-$ is the decrease in area. If $S'$ is positive, it indicates an overall seaward migration of the coastal zone area, and if $S'$ is negative, it indicates an overall landward migration of the coastal area.

Based on the coastlines of two different years, the symmetrical difference tool in ArcGIS was used to obtain the sea-land pattern for a certain time period.

### 3.3.3. Index of Coastline Type Diversity (ICTD)

The diversity of coastline development patterns can be captured by the index of coastline type diversity using the following equation [38]:

$$ICTD = 1 - \frac{\sum_{i}^{n} L_i^2}{\left(\sum_{i}^{n} L_i\right)^2} \tag{3}$$

where ICTD represents the Index of Coastline Type Diversity, $L_i$ is the length of the $i$th type of coastline, and n is the number of coastline types. An ICTD close to 0 indicates a single coastline type and low diversity in the region. ICTD values close to 1 indicate that the coastline types in the region are more complex and the diversity of each type is high.

Alternatively, the diversity of coastline types is also low when the structural tendency of coastline types in the region is more obvious, i.e., when the percentage of the length of one type of coastline type is significantly greater than that of other types. Therefore, coastline type diversity is low when there are fewer coastline types in the region.

### 3.3.4. Fractal Dimensions of Coastline

Common methods for calculating the fractal dimension of the coastline are the quantifier method and the grid method. Relevant studies have proven that the difference in the fractal dimensions of the coastline measured by the grid method is less than that of the quantifier method at different scales [40,41].

The basic idea of the grid method is to use square grids of different lengths to cover the measured coastline continuously and without overlapping. When the length of the square grid changes, the number of grids required to cover the entire coastline will change accordingly. According to fractal theory,

$$N_k(\varepsilon_k) \propto \varepsilon_k^{-D} \tag{4}$$

Taking the logarithm on both sides of Formula (4), the following is obtained:

$$lgN_k(\varepsilon_k) = -Dlg\varepsilon_k + A \tag{5}$$

where *A* is the pending constant and *D* is the fractal dimension of the measured coastline. The theoretical value of the fractal dimension of a straight line is 1, the theoretical value of the fractal dimension of a rectangle is 2, and the larger the fractal dimension value, the more complex the shape of the coastline indicated.

## 4. Results

This study used the visual interpretation method to extract and classify the coastline of mainland China. Figure 2 shows the distribution of coastlines for mainland China and four key regions for periods from 1990 to 2020. The study further analyzes the changing characteristics of the Chinese coastline from 1990 to 2020 through the coastline's change in length and types, the variation characteristics of land and ocean pattern, the index of coastline diversity, and the fractal dimensions of the coastline.

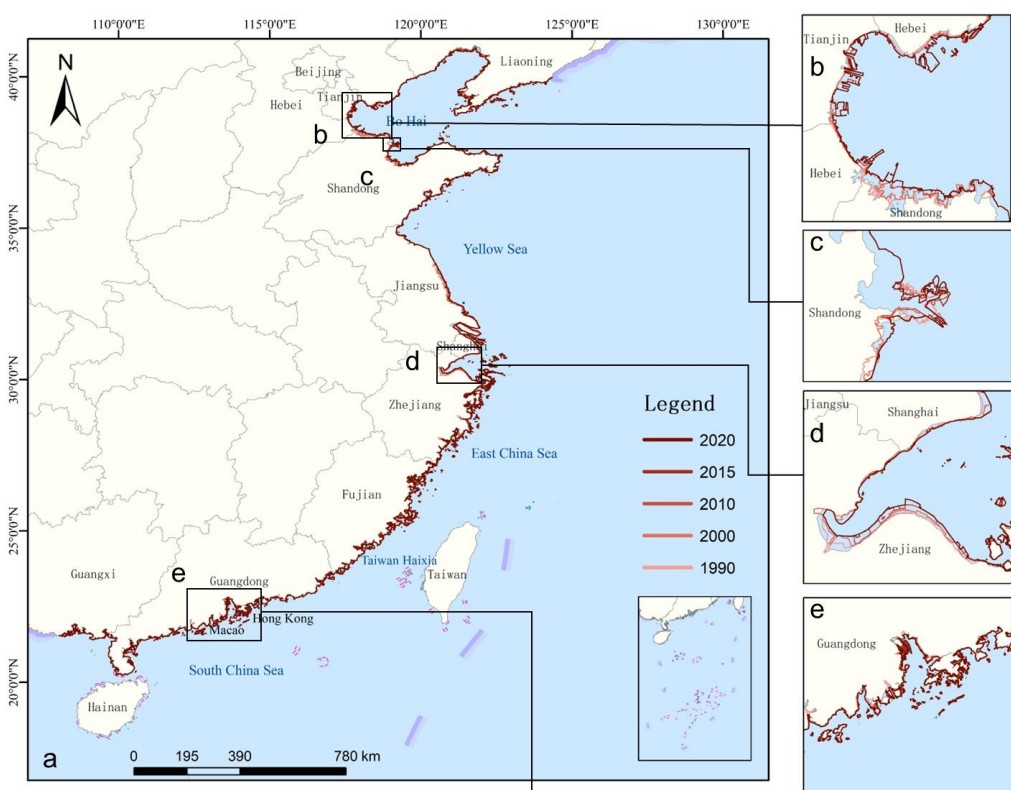

**Figure 2.** Distribution of coastline for mainland China (**a**), Bohai Bay (**b**), Yellow River Delta (**c**), Hangzhou Bay, (**d**) and Pearl River Delta (**e**) from 1990 to 2020.

### 4.1. Changes in Coastline Length and Types

Figure 3 illustrates the temporal variation of the total coastline length and the artificial and natural coastline lengths of mainland China from 1990 to 2020. During the past 30 years, the total coastline length of mainland China has increased from 30,041.22 km in 1990 to 32,977.34 km in 2020, an increase of 2936.12 km with an annual change rate of 97.87 km/year. Due to the rapid development of the coastal economy, the length of the artificial coastline increased rapidly from 10,022.49 km in 1990 to 17,660.84 km in 2020, an increase of about 80%, and exceeded the length of the natural coastline between 2010 and 2015. The length of the natural coastline decreased from 20,018.73 km to 15,316.5 km and was largely transformed into an artificial coastline.

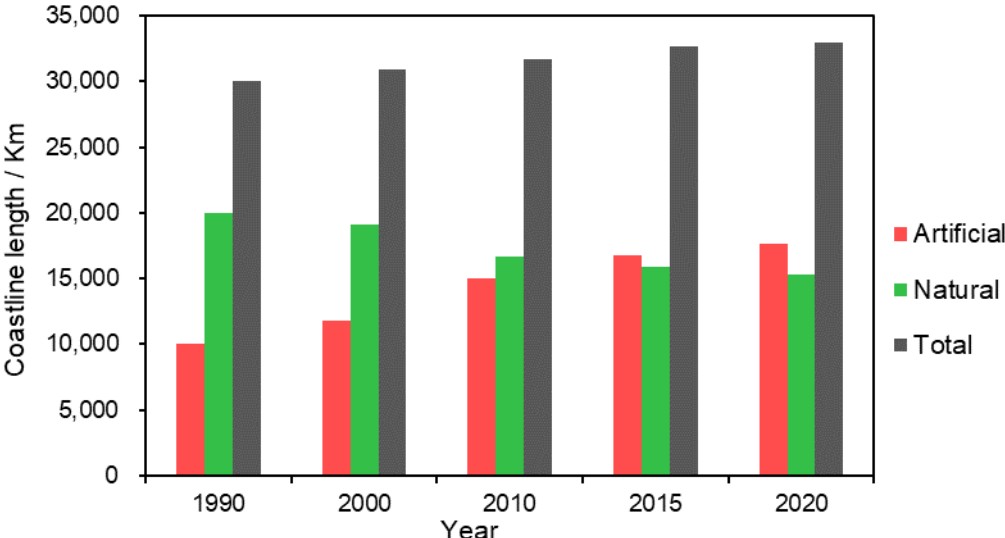

**Figure 3.** Total artificial and natural coastline length from 1990 to 2020.

Figure 4 shows the percentage of each type of coastline and the ICUD for mainland China for the period from 1990 to 2020. During this period, the ICUD in mainland China increased, and the proportion of artificial coastline increased significantly (Figure 4a). The results show that the ICUD increased the most between 2000 and 2010, at 8.37%. The proportion of artificial coastline increased from 33.32% in 1990 to 57.71% in 2020. The remaining coastline types show a decreasing trend, with the sandy coastline showing the largest decrease, accounting for a 7.3% decrease over the last three decades. It is followed by bedrock, which decreased from 17.54% to 10.47%.

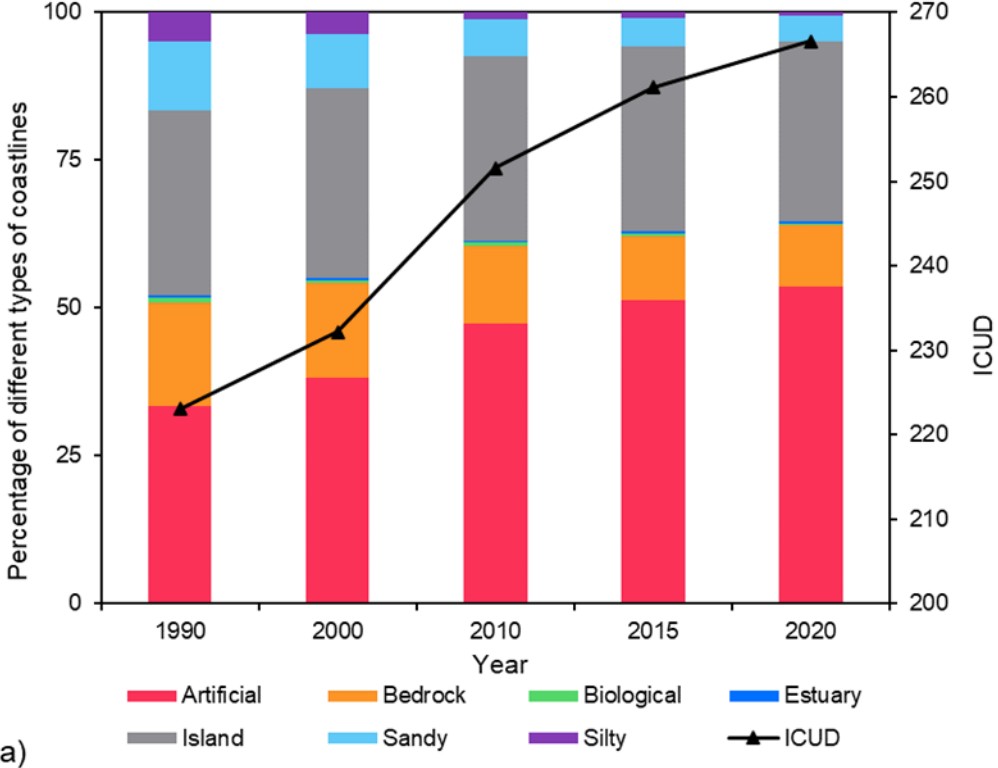

(a)

**Figure 4.** *Cont.*

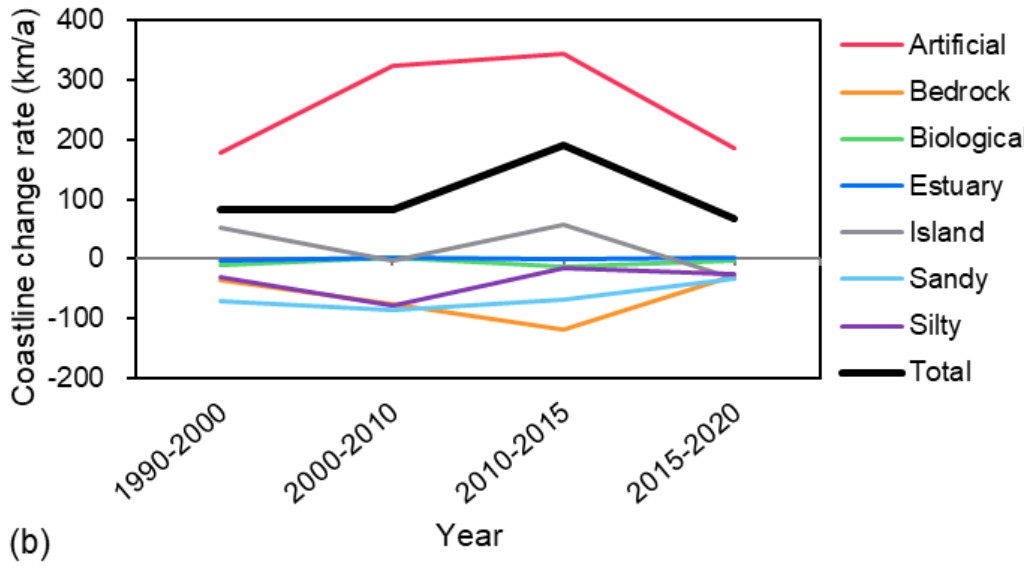

**Figure 4.** (**a**) Percentage of different types of coastlines and the ICUD from 1990 to 2020; (**b**) Change rate of different types of coastline from 1990 to 2020.

The change rate of total and artificial coastline lengths in mainland China displayed a trend of increasing and then decreasing, and the trend of change in coastline length of each type of natural coastline was more complex (Figure 4b). The change rate of total and artificial coastline lengths in China reached its maximum value between 2010 and 2015, at 189.48 km/year and 344.14 km/year, respectively. On the contrary, the change rate of bedrock coastline length reached the minimum value of −116.63 km/year during this period, indicating that human activities became more frequent in the coastal zone of mainland China during 2010–2015, resulting in significant changes to the coastline.

Figure 5 shows the coastline length of the eight provinces and four cities from 1990 to 2020. Along with the conversion of the coastline into an artificial coastline, most of the area has been straightened out. Zhejiang is the only region among the 12 provinces (cities) where the length of the coastline has decreased. It has decreased by 172.04 km in 30 years, but the coastline length is still the longest, at 6781.2 km. The length of coastlines in the remaining eleven provinces (cities) has increased over the past 30 years. Liaoning is the region with the largest increase in coastline length between 1990 and 2020, from 3199.53 km to 3908.61 km, an increase of 709.08 km. Macao is the region with the lowest increase in coastline length over the past 30 years, at only 15.15 km.

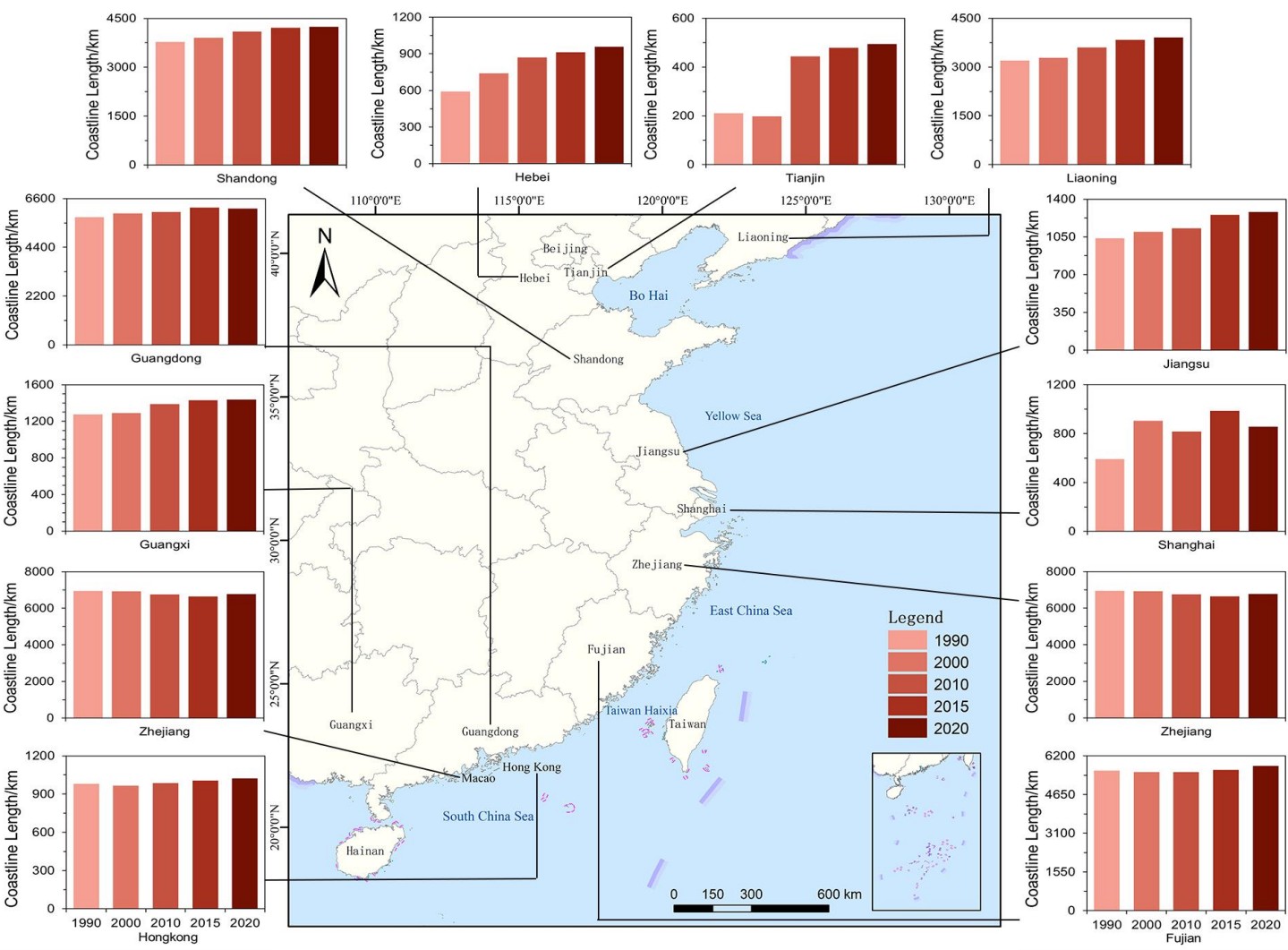

**Figure 5.** Coastline length of provinces in China from 1990 to 2020.

Table 2 lists the percentage change of coastline length and coastline change rate in 12 provinces of China's coastal region from 1990 to 2020. Tianjin has increased its coastline length over the past 30 years by about 135.00% of the 1990 level, with a rate of change of 9.46 km/year. Fujian is the region with the smallest increase in coastline length, with only 3.19% of the 1990 level and an average annual increase of 5.98 km. Although the percentage change of coastline length in Liaoning is not impressive, at 22.16%, it is the city with the largest coastline change rate, with a rate of change of 23.64 km/year.

**Table 2.** Percentage change of coastline length and coastline change rate in 12 provinces (cities) from 1990 to 2020.

| Province/City | Percentage Change of Coastline Length (%) | Coastline Change Rate (km/Year) | Province/City | Percentage Change of Coastline Length (%) | Coastline Change Rate (km/Year) |
|---|---|---|---|---|---|
| Liaoning | 22.16 | **23.64** | Zhejiang | −2.47 | −5.73 |
| Hebei | 61.64 | 12.16 | Fujian | 3.19 | 5.98 |
| Tianjin | **135.00** | 9.46 | Guangdong | 6.64 | 12.75 |
| Shandong | 12.16 | 15.31 | Hong Kong | 4.39 | 1.44 |
| Jiangsu | 23.59 | 8.15 | Macao | 35.27 | 0.51 |
| Shanghai | 44.38 | 8.77 | Guangxi | 12.82 | 5.45 |

*4.2. Variation Characteristics of Sea–Land Pattern*

Coastal erosion, estuarine siltation, and reclamation lead to changes in local and regional coastlines, resulting in changes in land and ocean patterns at macroscopic scales. Comparing continental coastlines in different periods can clearly reveal the characteristics of changes in sea–land patterns and their driving causes. The spatial area enclosed by the two phases of continental coastlines reflects the area scale characteristics of sea–land pattern changes; the spatial location of the coastlines at the beginning and end of the two phases reflects the directional characteristics of land advancing and sea retreating, or vice versa. The difference in the types of coastlines at the beginning and end of the two phases further reflects the driving causes of sea–land pattern changes.

As shown in Figures 6 and 7, the sea–land pattern of the mainland China coastal zone has been relatively homogeneous over the past 30 years, with most areas showing land advancing and the sea retreating and a few areas showing the opposite. Among them, the areas that experienced land advances and sea retreats are mainly distributed in Jiangsu Province, Hebei Province, and Tianjin city, north of Hangzhou Bay. The coastal zone of mainland China has seen a significant increase in area, with a net increase of about 10,902.55 km$^2$. Among the 12 provinces (or cities), the net increase in land area of the coastal zone of Shandong, Jiangsu, and Zhejiang provinces was the most significant, with a net increase of 2358.42 km$^2$, 2222.29 km$^2$, and 1373.84 km$^2$, respectively, accounting for 21.63%, 20.38%, and 12.60% of the net increase in the coastal zone of mainland China. The net increase in the coastal zone of Macao was the least, only 14.48 km$^2$.

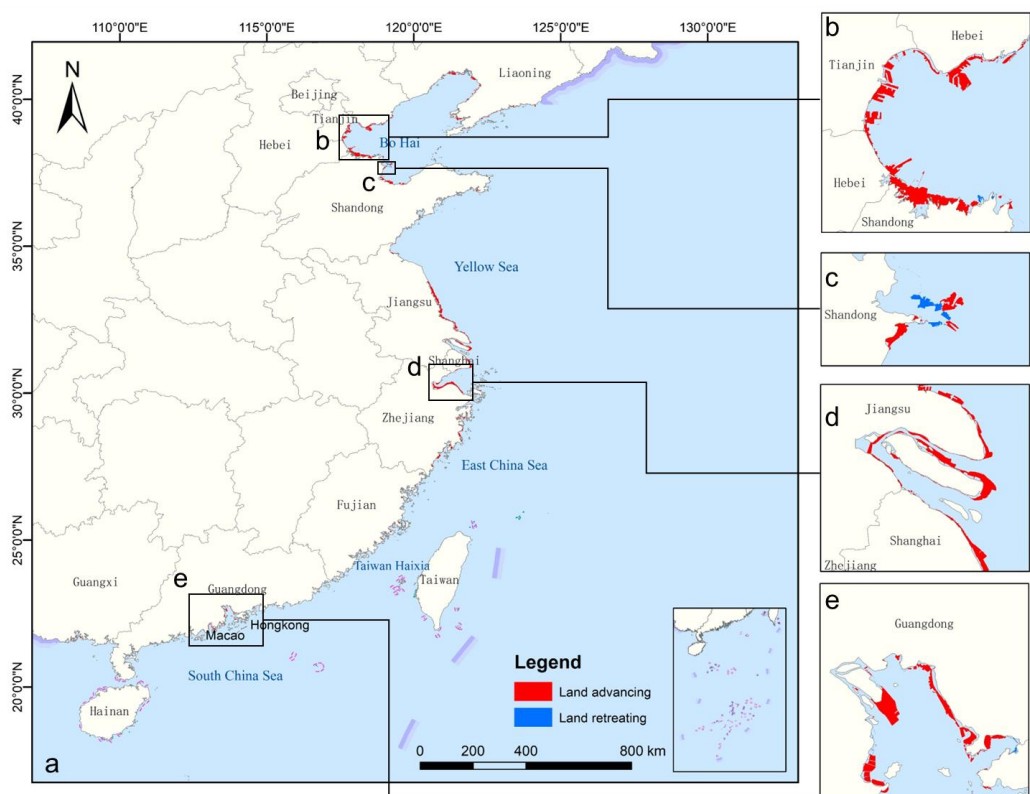

**Figure 6.** Spatial distribution for sea–land pattern in mainland China (**a**), Bohai Bay (**b**), Yellow River Delta (**c**), Hangzhou Bay, (**d**) and Pearl River Delta (**e**) from 1990 to 2020.

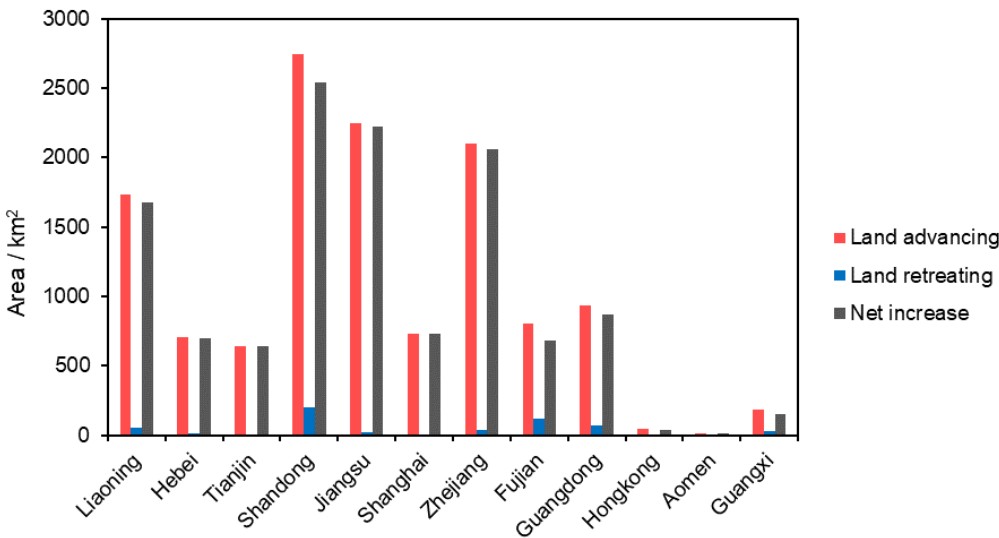

**Figure 7.** Area change of sea–land pattern from 1990 to 2020.

### 4.3. Index of Coastline Type Diversity (ICTD)

Figure 8 shows that the ICTD of most provinces and cities has decreased over the 30-year period, with the ICTD of Guangxi, Macao, and Jiangsu decreasing the most, falling by 0.286, 0.270, and 0.268, respectively. The structure of coastline types is beginning to lean toward artificial coastlines. Shanghai and Hong Kong are the only two regions with an increase in ICTD, rising by 0.044 and 0.013, respectively.

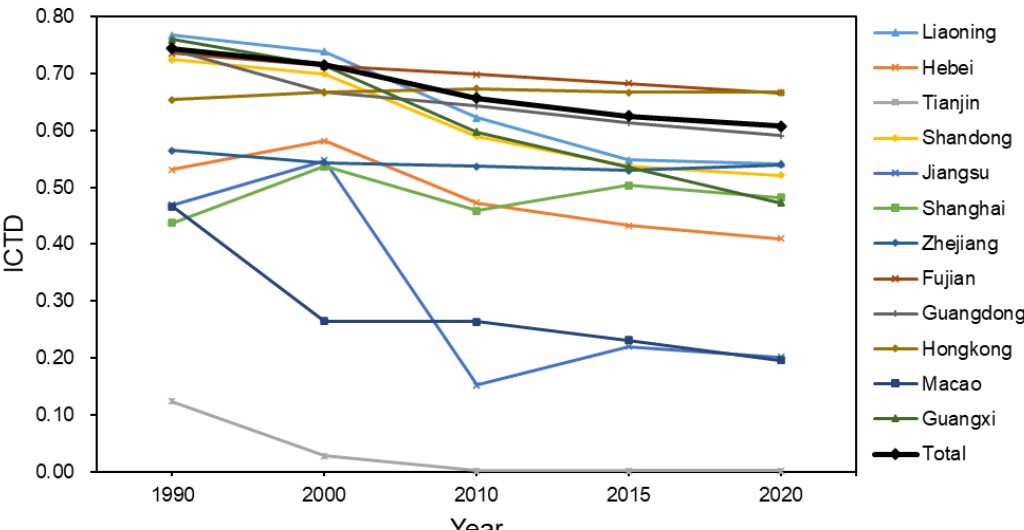

**Figure 8.** ICTD changes in 12 provinces and cities of China from 1990 to 2020.

The ICTD of Tianjin was the lowest during the 30 years, decreasing from 0.124 to 0.002, with a single coastline type; the ICTD of Hong Kong increased from 0.654 to 0.667, the largest among the 12 provinces and cities; and the ICTD of Fujian is similar, with the two regions having the most abundant coastline types and insignificant structural tendencies. The decreased ICTD indicates the coastlines in mainland China were converted into artificial coastline, and the coastline types that tended to be homogenized with the proportion of artificial coastlines have increased significantly.

### 4.4. Spatiotemporal Variations of Fractal Dimensions of Coastline

In order to analyze the changing characteristics of coastline morphology over a long period, this paper selected 11 scales of 600, 900, 1050, 1200, 1500, 1800, 3000, 4500, 6000, 7500, and 9000 to calculate the fractal dimension of the mainland China coastline in different spatial units by using a grid method (Figure 9).

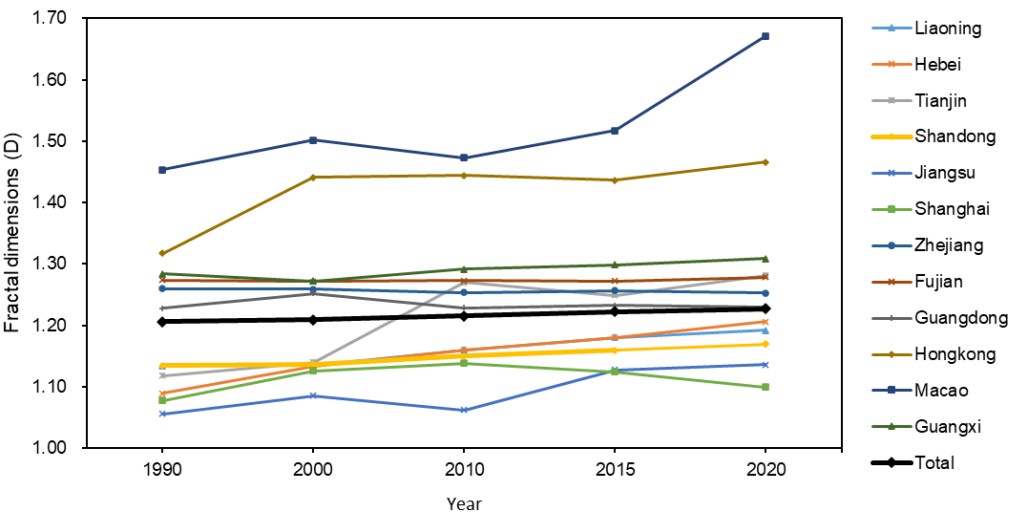

**Figure 9.** Fractal dimension changes of China coastline from 1990 to 2020.

The overall fractal dimension of the coastline of mainland China during 1990–2020 shows a continuous slow growth, indicating that the shape of the coastline tends to be complex. Among the 12 coastal provinces, the fractal dimension of Zhejiang, Guangdong, Hong Kong, Macao, Guangxi, and Fujian was always higher than that of the entire coastline

in the past 30 years. However, the fractal dimension of Liaoning, Hebei, Shandong, Jiangsu, and Shanghai is always lower than that of the entire coastline. From 2000 to 2010, the fractal dimension of Tianjin increased rapidly and exceeded the fractal dimension of the whole coastline. It was observed that the fractal dimension of Tianjin's coastline changed dramatically during the decade. In the past 30 years, the fractal dimension of the coastline of Hong Kong and Macao has always been higher than that of other provinces and cities; that is, the complexity is the highest.

## 5. Discussion

### 5.1. Impact of Changes on China's Coastline

Human interventions weaken the coastline's natural structure, causing it to become more vulnerable to the extreme climate conditions and inducing coastline erosion, significant reductions in mangrove area, and sediment deposition [24]. Since China's reform and opening up, coastal development patterns have changed significantly, with human development activities becoming the dominant factor in coastal change [6]. Though China's coastal provinces and cities generate more than 60% of the country's GDP, the varying degrees of development in each region have led to differences in the impact of human activities on the coastline's structure.

According to previous research [5,6,42] and our statistical data, in the last 30 years, the construction of wharves in the port area and some aquiculture areas on the protruding original coast have led to coastline growth due to the effects of human development. For example, the length of the artificial coastline increased from 10,022.49 km in 1990 to 17,660.84 km in 2020. Alternatively, natural factors such as sediment deposition in the estuary and the growth of sand dams have also increased the total length of the coastline; for example, the length of mainland China's coastline increased from 30,041.22 km in 1990 to 32,977.34 km in 2020.

The structure of China's mainland coastline shows more complex patterns and processing characteristics. However, the general trend indicates that the proportion of artificial coastlines has increased significantly, especially in the plain coasts and economically developed and densely populated estuaries, such as Tianjin (99% in 2020), Jiangsu (89% in 2020), Shanghai (60% in 2020), and Guangdong (55% in 2020), where artificial coastlines have increased dramatically over a 30-year period. Among natural coastlines, all types of coastlines have declined due to human activities, such as sea enclosing land, reclamation, port construction, and artificial farming. Bedrock coastlines are mainly distributed in the Liaodong Peninsula and Shandong Peninsula; silty coastlines are mostly concentrated in Bohai Bay and the Yellow River estuary; and biological coastlines are mostly distributed in coastal areas of Guangxi and Guangdong. The lengths of the sandy coastline and bedrock coastline decreased the most, at 2062.95 km and 1815.8 km, respectively.

During 1990–2020, China's coastal areas were influenced by policy, socio-economic, and natural environmental factors, with a large influx of people and frequent development and construction activities. As a result, the mainland coastline as a whole expands seaward, and the proportion of artificial coastline grows substantially.

By 1990, China had established special economic zones such as Shenzhen, Zhuhai, Shantou, and Xiamen, as well as economic open zones such as the Yangtze River Estuary and the Pearl River Estuary [43]. Since then, coastal economic open zones have been formed. In 2001, China formally joined the WTO, and the total tariff level dropped significantly, opening up coastal economic zones to the outside world and entering a new stage [44]. In 2013, China put forward the "One Belt, One Road" initiative, and the continuous improvement of the opening-up strategy to the outside world provided a convenient method for enterprises to conduct business overseas, and the quality of economic development in coastal areas continued to grow steadily [45]. These policies have led to an increase in the length of mainland China's coastline and in the proportion of artificial coastline from 1990 to 2015, with the growth rate of mainland China's coastline being the highest in 2010–2015 (189.48 km/year).

In the last 30 years, all coastal provinces and cities in China have experienced large economic development, but this has also put enormous pressure on the ecological environment of the coastal zone. These pressures are specifically manifested in the reduction in coastal wetlands and mangrove areas and the increase in coastal aquaculture areas. By 2020, the land surface area of China's coastal areas, with mangroves growing constantly, will only represent 18% of the mangrove area in 1973 [46]. The coastal wetland area decreased by 3288 km$^2$ between 2000 and 2010 [5]. The total area of aquaculture ponds increased from 2606.62 km$^2$ in 1985 to 12,099.52 km$^2$ in 2010 [47]. The trend of shrinkage in the area of coastal bays in China between 2010 and 2020 was significant, and the rate of shrinkage slowed significantly after 2015 [48]. Moreover, the reduction in coastal wetlands and mangrove areas may lead to a coastal zone ecosystem with a decreasing number of species and a decreasing amount of blue carbon storage. For example, reclamation projects in Hangzhou and Shenzhen have reduced the area of wetlands and mangroves, which in turn has led to a dramatic decline in bird species. The mangrove area in Beihai, Guangxi, was reduced by 150 ha between 1998 and 2003 due to activities such as deforestation and ponding, resulting in a loss of 82,392 tons of carbon stock during this time period [49].

In 2015, "Transforming our World: The 2030 Agenda for Sustainable Development" was officially adopted at the UN Sustainable Development Summit. The agenda included a set of Sustainable Development Goals (SDGs). Of these, SDG14.5 states that, by 2020, our world should conserve at least 10 percent of coastal and marine areas, consistent with national and international law and based on the best available scientific information. In response to SDG 14.5, China introduced the *Management Measures for Coastline Protection and Utilization* in 2017. These measures require the implementation of categorized protection and utilization of coastlines, which are divided into three categories: strict protection, restricted development, and optimized utilization, according to the condition of the natural resources of the coastline and the degree of development. It is proposed that, by 2020, the national natural coastline retention rate will be no less than 35%. Some provinces and cities are also targeting the implementation of pond return to the forests, reclamation control, and coastline ecological protection policies according to the situation of the coastal zone. It is believed that we will soon see the effect of these measures on the protection of the coastline.

*5.2. Regional Differences in Coastline Changes*

Temporal and spatial changes of the coastline also reflect regional differences and stage characteristics of coastal economic and social development. From 1990 to 2000, the economic development level of coastal provinces and cities was different from that of industrialization and urbanization, and the development level of southeastern coastal areas was generally higher than that of northern coastal areas. The coastline length of Guangdong, Shanghai, Shandong, and Jiangsu on the southeast coast has increased significantly, and the coastline length of Hebei on the north coast has also increased significantly. From 2000 to 2010, all coastal regions developed, and the regional gap gradually narrowed overall. Shanghai, Guangdong, and Zhejiang are in the first tier. In the developed areas where coastal cities are located, strong economic and social development has begun to influence and radiate to the surrounding areas. In most provinces and cities, the fractal dimension of the coastline and the complexity of the coastline increase, among which Tianjin has the largest increase. The coastline diversity index generally decreased; the coastline type tended to be simple.

Due to the influence of economic policies as well as demographic development in each region, three provinces and cities, Shanghai, Tianjin, and Guangdong, had the highest level of population–economy coupling during 2010–2015, followed by Jiangsu, Fujian, and Shandong provinces [49]. It can be seen that the level of coastal development in the north was stronger than that in the south during these five years [46]. The transformation reflected in the coastline is that the coastline lengths of northern provinces and cities such as Shanghai, Tianjin, and Liaoning have all increased significantly, with Shanghai, Shandong, and Liaoning showing the largest increases. Against the background of the adoption of

the Administrative Measures on Coastline Protection and Utilization and the coastal zone protection and restoration projects carried out by provinces and municipalities, the rate of coastline changes in mainland China showed a decreasing trend compared with that from 2010 to 2015. Among them, the rate of coastline changes in provinces and cities north of Hangzhou Bay decreased more than that in provinces and cities south of Hangzhou Bay.

A previous study shows that the change pattern of the coastline length in Bohai Bay shows a decrease between 1990 and 2000 and an increase between 2000 and 2015, as well as a continuous decrease in the fractal dimension of the coastline, an increasing proportion of artificial coastline, and a continuous seaward expansion of the coastline [50,51]. This is consistent with our findings and maintains the same pattern of change during 2015–2020. The main reason for this change in the pattern of coastline in this region is marine engineering, such as reclamation and port construction.

Influenced by the flow path of the Yellow River to the sea, the amount of water and sand, and the amount of precipitation, the coastline of the Yellow River Delta changes frequently and rapidly, and most of the coastline is natural, with the estuary continuously extending seaward between 1990 and 2020, with landward erosion dominating the northern coastline and seaward expansion dominating the southern one [52,53]. Despite the intensification of human activities in the Yellow River Delta region, due to the relatively short impact time, natural factors are still the main cause of this change.

During 1990–2020, the south coast of Hangzhou Bay was dominated by seaward expansion, with almost no change in the north and a significant increase in the proportion of artificial coastline. This is consistent with the results of previous studies [54,55]. The combined effect of human factors and natural factors together has caused the above results, where the northern shore is more influenced by human activities and the natural coastline is gradually replaced by artificial dykes. The coastline's shape is not easily changed. In contrast, the degree of coastline curvature on the south coast is mainly influenced more by natural factors, and the coastline is more complex.

It was found that, during 1990–2020, in the Pearl River Delta, the coastline expanded seaward as a whole, and the ratio of total length to artificial coastline increased continuously, while the fractal dimension and diversity index of the coastline showed a decreasing trend, and the areas with greater changes were mainly concentrated in Dapeng Bay, Shenzhen, Nansha, Guangzhou, and Doumen, Zhuhai, which was consistent with the findings of Xia et al. [56–58]. This is mainly due to the construction of coastal engineering, beach reclamation, sea-enclosing land, port construction, and artificial farming.

### 5.3. Inadequacy of Coastline Extraction Methods

Commonly used methods for the automatic extraction of coastlines include threshold segmentation methods [59], edge detection operator methods [60], data mining methods [61], etc. Automatic extraction methods have greatly improved efficiency and are suitable for large-scale extraction [62]. However, the extraction results are prone to discontinuity in areas with complex coastline conditions, large topographic undulations, and spatial differences and cannot be used to identify most of the coastline types [7].

Therefore, this study extracted the coastlines of China's mainland in 1990, 2000, 2010, 2015, and 2020 based on the visual interpretation method using remote sensing images. It was found that the visual interpretation method has high accuracy in determining the location and classifying the type of coastline. In addition, due to the large area considered in this study, the method required a large number of images. This method was also affected by the cloudy and rainy conditions in southern China, which meant that many data points had to be analyzed to find those that met the extraction requirements. Therefore, future work will investigate automatic data screening methods and consider automatic extraction with manual correction for coastline extraction.

## 6. Conclusions

In the past 30 years, the length of mainland China's coastline has continued to increase significantly, increasing from 30,041.22 km in 1990 to 32,977.34 km in 2020. The proportion of natural coastline decreased from 66.64% to 46.45% as the degree of artificial coastline increased. During the past 30 years, the coastline of mainland China has displayed an overall trend of seaward expansion, and Bohai Bay, Hangzhou Bay, and the Pearl River Delta have shown significant seaward expansion due to the impact of reclamation projects. In the Yellow River Delta, due to the influence of sediment deposition and the frequent channel changes of the estuary, landward erosion is still obvious despite intense human activities in recent years. The ICTD showed a downward trend overall. Due to the influence of port construction, Tianjin had a single coastline type, and the ICTD was always at its lowest level over the past 30 years, and the index continued to decrease. The fractal dimension of the coastline showed a slowly increasing trend, but the change was not significant in the past 30 years. In Macao, the fractal dimension of the coastline was always the highest due to its small city area.

Since the first year of the SDGs (2015), China has implemented a series of initiatives to protect the coastal ecosystem, such as the *Sustainable Development Strategy for the East Asian Seas 2015* and the *Special Program for Mangrove Protection and Restoration*. Since 2015, the growth rate of the mainland coastline and artificial coastline length has started to slow down, which shows that China has achieved some success in the scientific management and planning of the coastline. However, the structure of China's coastline still faces the problems of a high proportion of artificial coastline, a decreasing ICTD, and a complex structure. In the future, it is still necessary to strengthen the monitoring and management of coastlines to ensure the scientific and sustainable development and utilization of coastal zones. In the future, we will investigate automatic data screening methods and consider automatic extraction with manual correction for coastline extraction.

**Author Contributions:** K.L. was responsible for the concept and methodology of the study, as well as writing the manuscript. L.Z. supported the development of the methodology and the design of the experiment and contributed to the review and editing of the manuscript. B.C. and L.L. supported the concept development and the design of the experiment. J.Z. and F.Y. supported the statistical and thematic graphs of the experimental results. All authors have read and agreed to the published version of the manuscript.

**Funding:** The authors are grateful for the support from the Strategic Priority Research Program of the Chinese Academy of Sciences (Grant No. XDA19030105).

**Data Availability Statement:** The data presented in this study are available on request from the corresponding author (L.Z.).

**Acknowledgments:** The authors appreciate the valuable comments and constructive suggestions from the anonymous referees and the editors who helped improve the manuscript.

**Conflicts of Interest:** The authors declare no conflict of interest.

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
