# Peer review of "Analysis of China’s Coastline Changes during 1990–2020"

_remotesensing, doi:10.3390/rs15040981_

Round 1

Reviewer 1 Report

This manuscript proposed the dynamic changes of coastline in China for 30 years, with clear principles, detailed experiments, and reliable results. Generally, it can meet the standards of Remote Sensing publishing. The authors are invited to address the following issues before finally accepting.

1.     Is the spatial resolution of Landsat data able to meet the accuracy standard of this study?

2.     The driving force factors and its analysis are suggested to be added.

Reviewer 2 Report

I read with interest the paper by Li Kaixin and colleagues. Their work focuses on the study of satellite images since 1990 to analyze coastline changes in China.

Overall, I think the data are well presented and explained, the results are supported by the data

I will limit to the following comments and suggestions

The introduction should be enriched with more references of other relevant studies

The discussion should also cite more references. The authors state information that needs to be backed by relevant bibliography.

My suggestion is that the paper should be accepted with revisions.

You may find in attachment the annotated pdf with comments for the authors.

Reviewer 3 Report

This paper analyses the evolution of the Chinese coastline using Landsat data across five periods. However, there is a major drawback in the description of the remote sensing analysis method, and the following aspects are suggested for improvement.

1. The title is remote sensing analysis, but the description of how to conduct remote sensing analysis is seriously lacking in the present paper. The authors should clearly explain how to use remote sensing for shoreline analysis methods.

2. The introduction is somewhat ambiguous. The reason may be that the authors have not focused on the key issues that the paper is addressing. The review section needs to be further improved. The novelty of the paper and what exactly is the key problem addressed need to be made clear.

3. Suggest adding a regional overview map in the second section.

4. The classification of shorelines is not comparable in the paper; for example, estuaries and islands are not exactly natural shorelines. More to the point, how to distinguish these types using remote sensing methods is of greater concern to readers.

5. The authors should analyse shoreline changes at the same time intervals they chose. Such analyses would be comparable. It is unclear why the authors need to add the year 2015.

6. Figure 5 is a bit interesting. However, the authors must clearly explain how Figure 5 was obtained. I failed to find anything relevant in the methods section.

7. The discussion section should be focused on the weakness of the method, and further analysis of the results. The existing discussion might be more appropriately included in the results.

Reviewer 4 Report

I read this report with great interest precisely for the reasons outlined in the introduction to the study.  China has one of the fastest growing economies in the world and 40% of the country's population is concentrated in the coastal zone -in which many of the fastest growing cities are located.  These circumstances require that careful monitoring takes place in order to insure that sustainable development will occur in coming years.  In my opinion, one of the most important reasons for this study as to do with the damage that major storms (typhoons) are sure to do to China's very long coastline.  Such storms will affect both the natural shore and artificial shores constructed for the purpose of increased protection.  Thus, this study, which shows changes during five intervals of time from 1990 to 2022, sets out a clear marker for comparisons going forward.

    The paper is well illustrated.  My criticism has only to do with the need for slight improvement of the English.  The use of articles (the and a) is problematic and needs review.  For example, under the "Methods" section in the second paragraph, the article "A" should be inserted in the sentence to begin: A natural coastline refers to...

Likewise, Appendix A requires a similar fix.  Definitions for Bedrock, Sandy, Biological and Silty require the article "A" at the start of the definition.  The last definition for the class Artificial should begin "An artificial coastline.

Specific Comments:

1.  The principal question addressed by the authors in their submission is the need to access changes in China's long coastline, where 40% of the country's population live in major cities on or very close to the shore.  Auxiliary issues treated in their analysis look at the expansion of artificial shores and the effect of population pressures on the ecosystems of the remaining natural shoreline.  

2.  Other authors have considered this topic, for example: Chen, J. 1997. The impact of sea-level rise in China's coastal areas and its disaster hazard evaluation. J. Coastal Research, 13: 925-930.  In this sense, the topic is not original, BUT it represents an important and timely update on the same central questions.  The J. Coastal Research is a leading English-language journal and it is very important that the MDPI journal "Remote Sensing" also should provide this significant update in English so that the world beyond China may be educated about the current situation.  

3.  What the current paper adds to the literature is its reliance on satellite imagery and its relevance as an updated look at the question.  

4.  In my opinion, the paper is well formulated and I do not see any necessary changes in methodology.  

5.  The paper is very well illustrated and the conclusions adequately represent the content of the paper.  

6.  The references provided in the paper are adquate.

Round 2

Reviewer 3 Report

The authors have made not many changes. I am sorry that most of the questions I asked did not receive a good response. In particular, points 2, 4 and 7. I hope the authors could respond positively. 

Author Response

Dear Reviewer,

Thank you very much for your suggestions on the article, which are of great help to the improvement of the article.

Please see the letter of response in attachment.

Best wishes,

Li Zhang
